# Influence of the Type of Cement on the Action of the Admixture Containing Aluminum Powder

**DOI:** 10.3390/ma14112927

**Published:** 2021-05-29

**Authors:** Justyna Kuziak, Kamil Zalegowski, Wioletta Jackiewicz-Rek, Emilia Stanisławek

**Affiliations:** Department of Building Materials Engineering, Warsaw University of Technology, 00-637 Warsaw, Poland; k.zalegowski@il.pw.edu.pl (K.Z.); w.jackiewicz-rek@il.pw.edu.pl (W.J.-R.); emiliastanislawek93@gmail.com (E.S.)

**Keywords:** aluminum powder, concrete expansion, Portland cement, ground granulated blast-furnace slag cement

## Abstract

The study of the effect of cement type on the action of an admixture increasing the volume of concrete (containing aluminum powder), used in amounts of 0.5–1.5% of cement mass, was presented. The tests were carried out on cement mortars with Portland (CEM I) and ground granulated blast-furnace slag cement (CEM III). The following tests were carried out for the tested mortars: the air content in fresh mortars, compressive strength, flexural strength, increase in mortar volume, bulk density, pore structure evaluation (by the computer image analysis method) and changes in the concentration of OH^−^ ions during the hydration of used cements. Differences in the action of the tested admixture depending on the cement used were found. To induce the expansion of CEM III mortars, a smaller amount of admixture is required than in the case of CEM I cement. Using the admixture in amounts above 1% of the cement mass causes cracks of mortars with CEM III cement due to slow hydrogen evolution, which occurs after mortar plasticity is lost. The use of an aluminum-containing admixture reduces the strength properties of the cement mortars, the effect being stronger in the case of CEM III cement. The influence of the sample molding time on the admixture action was also found.

## 1. Introduction

Commonly used cement composites shrink during binding, mainly as a result of water evaporation. Shrinkage is an undesirable phenomenon and may result in the formation of cracks in concrete structures. The size of the shrinkage depends on the type of cement and on concrete hardening conditions [1], among other factors. It is possible to obtain non-shrinkage and even expansive concretes, i.e., those whose volume increases during setting and the first few days of hardening. For this purpose, expansive cements or admixtures increasing the volume of concrete are used.

Expansive cements contain calcium sulphate (increased content in relation to non-expansive cements) and Klein complex (C_4_A_3_S^−^; sinter) or calcium aluminate cements, or increased C_3_A content as an expansive agent [1,2,3]. During the cement setting, calcium sulfate reacts with aluminates (C_4_A_3_S^−^ or aluminates in calcium aluminate cement or C_3_A) to give ettringite, which crystallizes with increasing volume, causing the concrete to expand [3]. Expansive cements are used mainly for concrete shrinkage compensation [1,3], and the expansion of expansive cement paste up to 5% can be reached [4].

Expansive concretes made of expansive cements are characterized by lower total porosity and thus lower permeability of composite, as well as better frost resistance and higher strength compared to concretes from non-expansive cements [5]. The disadvantage of sulfate-containing expansive cements is the possibility of the formation of ettringite after the cement sets, which leads to cracking of the cement matrix.

Another way to cause concrete expansion is to use concrete admixtures. Admixtures increasing the volume of the concrete mix can be divided into expansive, gas-liberating, and foaming [6].

Expansive admixtures for the production of expansive concretes are calcium sulfates and aluminates (components of expansive cements), as well as calcium oxide and magnesium oxide [3], which in reaction with water give calcium hydroxide and magnesium hydroxide, respectively, which crystallize with increasing volume. Expansive admixtures are used for the production of injection preparations and repair mortars [6]. According to European standard EN 934-4 [7], expanding grout admixtures are used to ensure that the volume of the grout does not decrease during hardening. The increase in the volume of the grout during this time should not exceed 5%. 

Hydrophobic organic surface-active compounds, e.g., sodium or potassium abietate, as well as proteins of animal origin are used as foaming admixtures. When mixing the grout with such an admixture, abundant foam is created, which increases the volume of the grout. Foaming admixtures are mainly used for the production of insulation materials [6,8].

Powders of aluminum, zinc, or silicon, as well as sodium borohydride or hydrogen peroxide, are used as gas-liberating agents. Gas-liberating agents are recommended to obtain the expansive cement pastes used in the filling of ducts in post-tensioned prestressed concrete, injection preparations, and in repair mortars and mortars for anchoring. Gas-liberating agents (mainly aluminum powder) are also used for the production of aerated concrete [9,10,11] and porous geopolymers [12,13,14]. The expanding effect of aluminum, zinc and silicon comes from the reaction of these elements with Ca(OH)_2_ created during cement setting. In the case of aluminum, this process occurs according to the chemical reaction [1,15]:2Al + 3Ca(OH)_2_ + 6H_2_O →3CaO · Al_2_O_3_ · 6H_2_O + 3H_2_(1)
2Al + Ca(OH)_2_ + 6H_2_O → Ca[Al(OH)_4_]_2_ + 3H_2_(2)

The presence of gibbsite, Al(OH)_3_ was also found as a reaction product of aluminum with components of the cement paste in the Portland cement paste [16]. 

The borohydride reacts with water to form hydrogen.
NaBH_4_ + 2H_2_O = NaBO_2_ + 4H_2_(3)

Hydrogen peroxide is disproportionated, and the product is oxygen gas.
2H_2_O_2_ = 2H_2_O + O_2_(4)

As a result of this reaction, gas is released, which creates many fine gas bubbles. They have an expanding effect on the concrete mix that has not yet set, thus counteracting the shrinkage of the resulting materials.

The most commonly used gas-liberating agent in concrete is aluminium powder. Hydrogen released in the reaction of aluminum with calcium hydroxide causes the formation of pores in the size range of 0.1–1.0 mm in the concrete [9], which results in an increase in the porosity of the concrete [9,10,17,18,19]. However, no significant changes in the content of micropores smaller than 1 µm due to the action of aluminum powder were reported. The addition of aluminum to the cement grout causes an increase in the macropores content [17]. With the increase in porosity, due to the action of aluminum powder, a decrease in the dry thermal conductivity was also observed [10,17,19]. Increasing the amount of aluminum powder results in increased porosity, decreased density [9,20], a decrease in modulus of elasticity [21], and usually decreased compressive strength [9,22]. In some studies, a uniform decrease in compressive strength with increasing aluminum content was not observed. Liu et al., with increasing aluminum powder content, observed first an increase in compressive strength, then a decrease, and then an increase again, but the compressive strength was only higher than that of cement paste without aluminium addition for 0.01% aluminum powder [23]. Such changes may be related to the shape and orientation of the pores. Elliptical pores, which can form due to hydrogen evolution, can result in mechanical anisotropy of the porous material [23,24]. 

Aluminum powder also affects the properties of fresh cement grout: it extends setting time, reduces water bleeding, and acts as a plasticizer [22]. It was also found that the addition of aluminum powder increases the bond strength anchor-grout interface [22]. 

The effect of aluminum powder on the properties of the mortar depends, among others on its fineness. With the increase in aluminum powder fineness, the density and compressive strength of the cement mortars decreased [18]. The reactivity of the aluminum powder also depends strongly on the pH of the environment, which depends on the type of binder used. Aluminum is less reactive in grout with 20% Portland cement and 80% blast furnace slag than in Portland cement. In a study using aluminum powder with a size of 1 mm, the presence of metallic aluminum (unreacted aluminum) in hardened cement paste with 80% blast furnace slag was found. No metallic aluminum was found in the hardened Portland cement paste [16]. There are many articles on the use of aluminum powder as an admixture for concrete, but the literature data lack a comparison of the action of aluminum-containing admixtures in composites based on different cements.

The aim of the study presented in this paper was to assess the effect of cement type on the action of an admixture (containing aluminum powder) increasing the volume of concrete, used in an amount of 0.5–1.5% of cement mass. Selected properties of cement mortars with Portland and ground granulated blast-furnace slag cement were tested.

## 2. Materials and Methods

The tests were carried out on cement mortars with dimensions of 40 × 40 × 160 mm^3^ differing in the type of cement and the amount of gas-liberating agent. Portland cement CEM I 42.5R (CEM I, Lafarge, Małogoszcz, Poland) and ground granulated blast-furnace slag cement CEM III/A 42.5N LH HSR NA (CEM III, Górażdże Cement SA, Chorula, Poland) were used. The mortar w/c ratio was 0.5. Quartz sand meeting the requirements of European standard PN-EN 196-1: 2006 [25] was used as aggregate. The cement/sand ratio was 1/3. As a gas-liberating agent, an admixture containing metallic aluminum and a plasticizer were used. The admixture doses of 0%, 0.5%, 1%, and 1.5% of cement mass (% m.c.) was mixed with cement before mixing the cement with water. Samples were made in accordance with the EN 480-1 [26] standard. Some samples were formed immediately after mixing the mortar components and some samples were formed after determining the air content in the mortar. Samples of mortars after demolding were cured in water. Mortar markings are presented in Table 1.

The scope of the tests included the following determinations:For fresh mortars:-Air content in the mortars according to EN 1015-7: 1998 [27]. The determinations were made after approx. 12 min from cement contact with water and additionally for CEM III 1.0% and CEM III 1.5% after 30 and 45 min from cement contact with water.
For hardened mortars: -Pore structure was evaluated by the computer image analysis method; the stereological method used for the quantitative description of three-dimensional phases in a material volume based on measurements performed on two-dimensional images of a material microstructure (images of sample cross-sections) [28,29]. The procedure of sample preparation for microstructure studies involved (1) cutting out one slice about 40 mm × 40 mm × 10 mm in size from a 40 mm × 40 mm × 160 mm sample for each concrete, (2) cold mounting under lowered pressure using a colored epoxy resin, (3) grinding, and (4) final polishing. After this, 2D images of the microstructure with a resolution of 800 DPI were taken using a computer scanner (EPSON, Bekasi, Indonesia). The images were subjected to computer processing in order to obtain the most precise binary image of black pores, which were of interest, on a white background of other microstructure constituents. The image processing was a combination of changes of contrast, brightness, color saturation, etc., followed by selection of pores and binarization. Quantitative analysis of the mortar’s microstructure was performed using a proprietary computer program working in a MATLAB environment which was developed at the Faculty of Civil Engineering of Warsaw University of Technology. This program was used to calculate the relative volume fraction of pores *V*_V_ and the relative surface area of pores *S*_V_.-Compressive strength and flexural strength according to EN 1015-11: 2001 [30] after 7, 28, 56 and 91 days of hardening of mortars formed immediately after mixing the mortar components and formed after air content tests. The flexural strength was determined by the 3-point bend test using mortar samples with dimensions of 40 × 40 × 160 mm^3^.-Bulk density.


## 3. Results and Discussion

### 3.1. Air Content in the Fresh Mortars

In the case of mortars with Portland cement CEM I, the content of air increased with the increase of the gas-liberating agent content (Figure 1). This effect is associated with the evolution of hydrogen gas in the mortars due to the reaction of aluminum from the admixture with OH^−^ ions formed during cement hydration. The increase in air content when using an admixture above 1% of CEM I cement mass was small.

In the case of mortars with CEM III cement, the addition of an admixture in the amount of 0.5% did not affect the result of the air content test. The increase of the admixture content to 1% of cement mass resulted in an increase in air content, but much smaller than in the case of mortars with Portland cement. The increase of the admixture content to 1.5% in mortars with CEM III cement did not increase the air content compared to the mortar with 1% admixture.

Due to the low impact of the admixture on the increase in air content in mixtures with CEM III cement, additional tests were performed after 30 and 45 min from the contact of water with cement. Their results, however, were close to the results after 12 min. Higher air content in the fresh mortar increases the porosity of the hardened cement composite, and thus reduces the strength of the mortar. Increasing the porosity also reduces the density of the material and improves its insulating properties.

### 3.2. Content of OH^−^ Ions in Cement Pastes

Smaller changes of air content test results for mortars made with CEM III cement may be associated with the lower alkalinity of the mortars in comparison to the mortars with Portland cement. The concentration of OH^−^ ions in the Portland cement paste quickly reached approx. 0.08 mol/dm^3^ (this content of OH^−^ corresponds to pH 12.9) and remained at this level throughout the test (Figure 2). The addition of an admixture containing aluminum powder to the cement paste caused a decrease in the concentration of OH^−^ ions to the level of about 0.065 mol/dm^3^, which indicates the reaction of OH^−^ ions with the aluminum from the admixture. The course of changes in OH^−^ concentration in the paste with admixture in time indicates a rather slow reaction at the initial cement setting time (a slight difference between the concentration of OH^−^ in the paste without and with admixture after 5 min of cement hydration).

The concentration of OH^−^ ions in the paste with CEM III cement increased more slowly than in the paste with CEM I cement. In the initial hydration period (3–5 min), it was more than two times lower than in the paste with CEM I. The concentration of OH^−^ ions in the paste with CEM III after an hour of hydration was about 0.06 mol/dm^3^ and was about 27% smaller than the OH^−^ content in the paste with CEM I cement. Furthermore, the difference between the concentration of OH^−^ ions in the paste with CEM III cement without admixture and with the admixture was smaller than in the case of CEM I cement. The chemical reaction rate depends on the concentration of reagents; therefore, with a lower concentration of OH^−^ ions in the CEM III paste, the reaction with aluminum and the evolution of hydrogen were slower compared to the paste with CEM I cement.

### 3.3. Expansion of Mortars

After the mortar samples were formed, a visible increase in the height of the bars containing the gas-liberating agent in the amounts of 1% and 1.5% of cement mass was observed (Table 2). As the content of the admixture containing aluminum powder increased, the amount of hydrogen gas (product of the reaction of aluminum and calcium hydroxide from the cement paste) increased. The hydrogen gas produced caused the mortar to swell (expand). CEM I cement-based mortars showed lower expansion with respect to CEM III cement mortars because the evolution of hydrogen gas was faster in the more alkaline mortars with CEM I cement. Part of the hydrogen gas produced in mortars with CEM I cement was probably lost as a result of volatilization from the fresh mortar during mixing and forming samples. The results of the air content test in fresh mortars (higher values in the case of mortar with CEM I) also indicate faster hydrogen release in mortars with CEM I. Mortars with CEM III cement showed greater expansion due to a slower reaction with aluminum (slower evolution of hydrogen). The expansion of mortar with CEM III cement proceeded more slowly, also after the loss of plasticity of the mix, which resulted in the cracking of samples with admixture containing 1 and 1.5% of cement mass (Figure 3). More cracks were observed on the top surface of the samples, as this is where the greatest stresses occurred during the increase in the volume of the samples. The lateral surfaces of the samples were delimited by the mold. In the case of Portland cement samples, the reaction of hydrogen evolution was faster and some of the hydrogen volatilized while the mixture remained in a plastic state. This resulted in less efficient sample expansion but no cracking of the samples. The gas-liberating agent used in the amount of 0.5% of cement mass did not increase the height of mortar samples with Portland cement.

### 3.4. Bulk Density of Mortars

As expected, as the content of the gas-liberating admixture increased, the mortars’ bulk density decreased (Table 3). The reason for the decrease in density is the “aeration” of samples with evolved hydrogen gas due to the reaction of aluminum powder with calcium hydroxide from cement paste. No significant effect of cement type on changes in the density of mortar with the admixture was observed.

### 3.5. Porosity of Mortars

The admixture containing aluminum powder added in an amount of 1 and 1.5% of cement mass visibly increased the porosity of the mortar (Figure 4 and Figure 5, Table 4). In CEM I cement mortars, evenly distributed spherical pores of varying diameter (up to approx. 2.5 mm) were observed. The number of pores increased with the amount of admixture. The porosity of the samples increased about 1.5 times after adding 1% admixture and more than 2-fold after using 1.5% admixture.

The porosity of the mortar with CEM III cement without admixture was about 40% lower than the porosity of the mortar with Portland cement. Addition of 1% of admixture caused an increase in the number of spherical pores of mortar with CEM III cement, which resulted in a more than 2-fold increase in mortar porosity. After increasing the amount of admixture to 1.5% of the CEM III cement mass, porosity increased more than four times (from 4.24% to 18.4%) compared to the mortar without admixture. Stratification of the sample of CEM III cement mortar with 1.5% of admixture was observed due to the pressure exerted by the hydrogen gas created after the loss of plasticity of the fresh mortar.

The increase in admixture content caused the increase in *S*_V_ of the mortar with both CEM I and CEM III (Table 3), which is associated with the appearance of new pores. The *S*_V_ parameter increased about 67% and 162%, respectively, after the addition of 1.0% and 1.5% of gas-liberating admixture to CEM I mortar. While the modification of CEM III mortar with the same amount of admixture resulted in an increase in *S*_V_ above 2-fold (about 227%) and about 5-fold (about 500%), respectively. The more intensive increase in *S*_V_ of CEM III mortar in comparison to CEM I mortar may be explained by a greater number of small pores introduced to slag cement mortar as a result of the modification. The results of the relative surface area of pores confirm findings concerning the relative volume of pores.

### 3.6. Compressive Strength of Mortar

The admixture containing aluminum powder used in an amount of 0.5% caused a reduction of approx. 8% in the compressive strength of mortars with CEM I cement after 7 days of hardening compared to the sample without admixture. After a longer hardening period, the decrease in compressive strength of CEM I 0.5% mortars compared to mortars without admixture was insignificant (Figure 6a). The increase of the amount of admixture to 1% cement mass reduced the compressive strength of mortars with CEM I cement by approximately 16%. The relationship between the compressive strength (*f*_c_) and the content of the admixture (*C*_A_) in the range of 0.5–1.5% of admixture was linear (Table 4). When the admixture was dosed at 1.5%, a slight decrease in strength was observed after the 28th day of hardening (the compressive strength after 56 and 91 days was lower than after 28 days of hardening). A similar effect was noted in the case of concrete testing with an admixture containing aluminum [31]. This may indicate that the aluminum reaction with calcium hydroxide was not finished in the initial period of mortar hardening and the evolution of hydrogen also after a longer ripening time, which may result in the formation of micro-cracks in the cement matrix. It is worth noting that the admixture used in quantities above 1 and 1.5% reduced the strength below the required for cement class 42.5 according to EN 197-1 [32].

In the case of mortars with CEM III cement, the admixture used in an amount of 0.5% of cement mass caused a significant reduction in the compressive strength of the mortars. The compressive strength of the mortars with CEM III cement decreased with the increase of the content of the admixture in the whole range of admixture dosing (Figure 6b). A linear relationship between the compressive strength and the content of the admixture with good correlation (R^2^ > 0.95) was obtained (Table 5). The compressive strength of mortars with CEM III cement decreased by approx. 16% for every 0.5% admixture. Slightly greater strength decreased the compressive strength observed as the content of the admixture increased after 91 days of hardening.

The compressive strength of the mortars with CEM III cement after 7 days was much lower than with CEM I, which is associated with a lower content of clinker in CEM III cement. After a longer hardening time, the mortars without admixture and with CEM III had a greater compressive strength than the CEM I 0% samples. The addition of admixture resulted in a greater reduction in the compressive strength of mortars with CEM III than with CEM I cement and the compressive strength of samples with admixture after 28 days and later were similar for both types of cement.

### 3.7. Flexural Strength of Mortars

The flexural strength of mortars with CEM III cement without admixture after 7 days was lower, and after 28, 56, and 90 days higher than the strength of mortars with CEM I cement without admixture. This is related to the cement composition and the rate of hydration of its components. In the case of composites with CEM III cement, which contains less Portland cement clinker (including fewer calcium silicates), fewer hydrated calcium silicates were produced in the early hardening period, and therefore the strength of mortars with CEM III cement after 7 days was lower than the mortar with CEM I cement. The components of the blast furnace slag present in CEM III hydrate more slowly but in a longer period of time than the components of the Portland cement clinker. The products of these reactions seal the cement matrix, which increases the strength of the composites. Therefore, composites with CEM III cement are characterized by a longer development of strength and a greater increase in strength after a longer hydration time compared to composites with CEM I cement and the strength of composites with CEM III after a longer hardening time was higher than that of composites with CEM I and the same composition [3,33]. The flexural strength of mortars based on both Portland cement and of ground granulated blast-furnace slag cement decreased when the content of the expansive admixture increased (Figure 7), while the changes in flexural strength were much greater for mortars with CEM III cement. The decrease in the flexural strength of the samples is related to the increase in their porosity due to the introduction of the admixture containing aluminum powder.

The addition of the admixture in the amount of 1.5% of cement mass reduced the flexural strength of CEM I-based mortars by 15%, and mortars with CEM III cement by 46% compared to the reference mortars. Mortars with CEM III cement showed an increase in flexural strength with the time of their hardening. For mortars with Portland cement, up to 28 days of hardening, flexural strength increased. Contrary to expectations, after 28 days, the values of flexural strength decreased for all compositions with CEM I. Mortars with CEM III cement obtained lower early flexural strength and higher flexural strength after 56 and 91 days of hardening, than mortars with Portland cement. A linear relationship between the flexural strength (*f*_f_) and the content of the admixture with good correlation was obtained (Table 6).

### 3.8. Influence of Sample Molding Time

Mortars with CEM I cement and with the admixture containing aluminum formed after testing the air content, i.e., after about 20 min from the contact of cement with water, showed a similar decrease in strength as mortars formed immediately after mixing the cement components (about 5 min after contact of cement with water) (Figure 8). The strength of the CEM III 0.5% mortar did not significantly depend on the molding time. On the other hand, adding more of the admixture to the mortar with CEM III resulted in smaller drops in strength in the case of later sample forming (after testing the air content, about 50 min from the contact of cement with water).

The influence of the sample forming time on the volume increase of the samples with an admixture of 1 and 1.5% was also noticed (Figure 9). The samples formed later showed a lower volume increase. Later forming of the samples may release some of the hydrogen from the fresh mortar as a result of longer mixing (transfer mortar from the measuring container to the mold), resulting in lower volume increase and lower drops in the porosity and strength of the samples. The smaller volume increases after the later sample forming were more noticeable with the CEM III cement. The smaller increase in the volume of samples with CEM III formed after a longer time after mixing the components also corresponds to less noticeable cracking of the samples. The differences noticed between the CEM I and CEM III cement samples formed at different times from mixing the mortar components result from the reactivity of cement in relation to the admixture with aluminum and the rate of hydrogen evolution. CEM I cement paste is characterized by a higher content of OH- ions and faster reaction with aluminum. Hydrogen is released mainly while the mortar with CEM I cement remains in a plastic state. Mortars with CEM III cement are less alkaline (lower content of OH^−^ ions) and the reaction with aluminum is slower. Hydrogen gas is also likely to be released after the mortar loses plasticity, which causes stress in the mortar and cracks it. Partial release of hydrogen as a result of mixing the mortar with CEM III after a longer time after mixing the components partially eliminated the unfavorable effect of the admixture used in larger amounts (decrease in strength and cracking of the sample).

## 4. Conclusions

The influence of the admixture containing aluminum powder on selected properties of cement mortars with CEM I and CEM III cement was investigated. The admixture was used in the amount of 0.5, 1.0, and 1.5% of the mass of the cement. Differences in the influence of the admixture on the tested properties of mortars depending on the type of cement used are listed below.

To induce the expansion of CEM III mortars, a smaller amount of admixture was required than in the case of CEM I cement. The aluminum-containing admixture caused a significant increase in the volume of CEM III mortars when dosed in the amount of 0.5%, while for CEM I mortars, a dosage above 0.5% was required to expand the samples.

An increase in the aluminum powder content caused a decrease in the bulk density and an increase in the porosity of cement composites. This effect was stronger in the case of mortars with CEM III cement.

The greater amount of tested admixture in the mixes increased the result of the air content test in the mixes with Portland cement due to the evolution of hydrogen in the reaction of aluminum with OH- ions. Mixtures based on CEM III cement exhibited slower hydrogen evolution, which resulted in lower changes in air content determinations while the mix remained plastic.

Caution should be exercised when using an admixture with aluminum and CEM III, because too much admixture (>1% of cement mass) caused cracks due to slow hydrogen evolution, which also occurred after plasticity loss and resulted in increased pressure inside the samples.

The use of aluminum-containing admixture reduced the strength properties of cement mortars, the effect being stronger in the case of CEM III cement. The compressive strength of composites decreased with increasing content of the expansive admixture, with a greater effect on mortars with ground granulated blast-furnace slag cement CEM III. In the case of mortars with CEM III and 1.5% admixture of cement weight, the compressive strength was lower by almost 50% compared to the control mortar. The flexural strength of composites decreased with the increase in the content of expansive admixture, while for mortars with CEM III cement, a greater influence of the admixture on the strength properties of the mortars was noted. In the case of mortars with CEM III and the admixture dosing of 1.5% of the cement mass, the flexural strength decreased by about 46% compared to the control mortar.

Delayed sample formation affects the admixture action. A smaller volume increase with later sample forming was observed. Later sample formation also resulted in lower drops in the strength of the mortar with CEM III cement.

## Figures and Tables

**Figure 1 materials-14-02927-f001:**
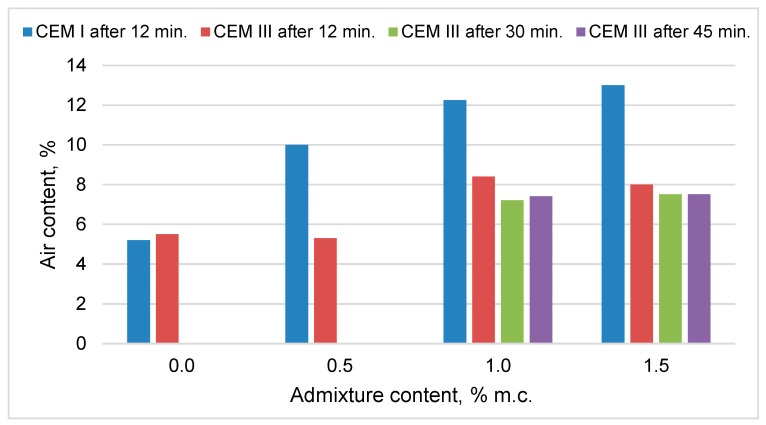
Comparison of test results for the air content in the mixture depending on the cement used and the content of admixture containing aluminum powder.

**Figure 2 materials-14-02927-f002:**
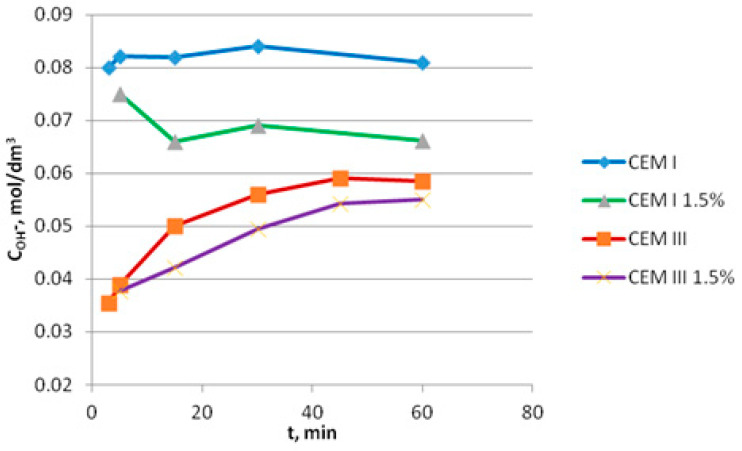
Changes in the OH^−^ ion concentration during hydration of CEM I and CEM III cements without and with the admixture containing aluminum powder.

**Figure 3 materials-14-02927-f003:**
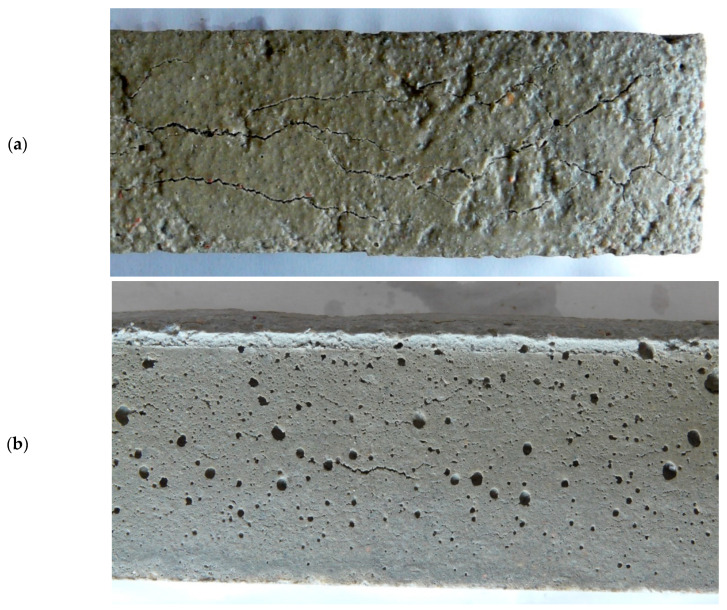
Cracks in the mortar sample with CEM III and gas-liberating admixture content equal to 1.5% of the cement mass (**a**) top surface, (**b**) lateral surface.

**Figure 4 materials-14-02927-f004:**
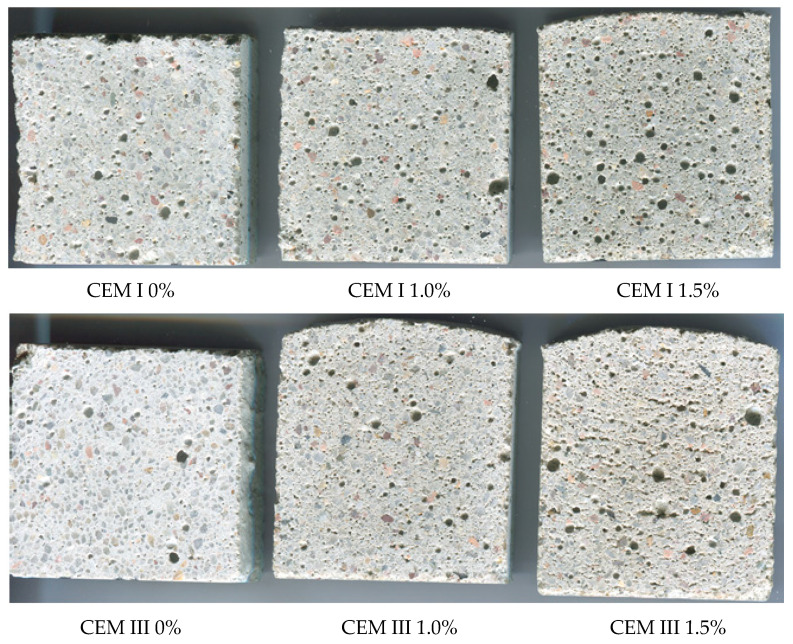
Images of sample cross-sections of mortars without and with the gas-liberating admixture.

**Figure 5 materials-14-02927-f005:**
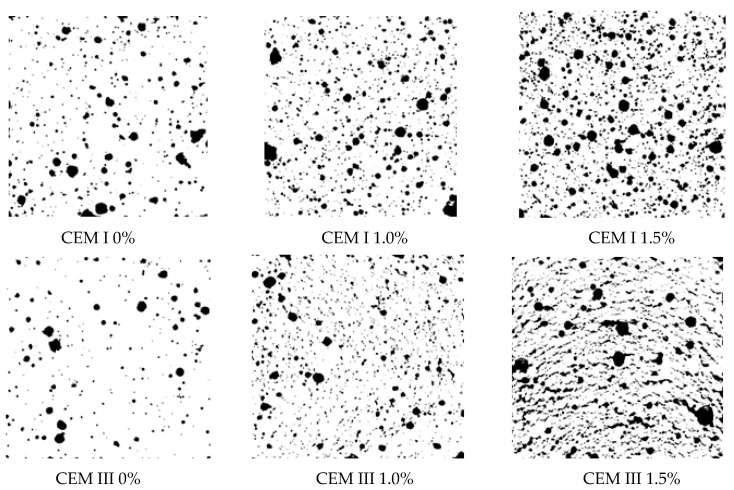
Images of sample cross-sections of mortars after preparation for image analysis.

**Figure 6 materials-14-02927-f006:**
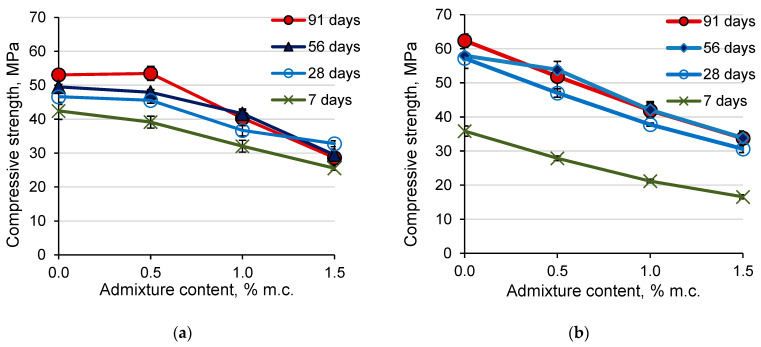
The effect of the admixture containing aluminum on the compressive strength of mortars with (**a**) CEM I cement, (**b**) CEM III cement.

**Figure 7 materials-14-02927-f007:**
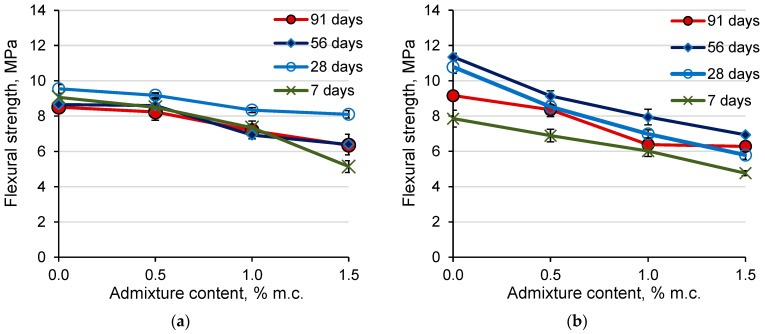
The effect of the admixture containing aluminum on the flexural strength of mortars with (**a**) CEM I cement, (**b**) CEM III cement.

**Figure 8 materials-14-02927-f008:**
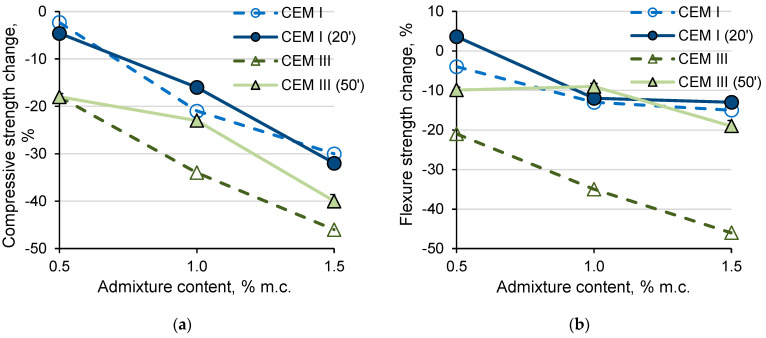
Compressive (**a**) and flexural (**b**) strength change of mortar with the admixture in relation to the mortar without admixture formed after mixing components of mortar (CEM I and CEM III) and formed after air content test (CEM I (20′) and CEM III (50′)).

**Figure 9 materials-14-02927-f009:**
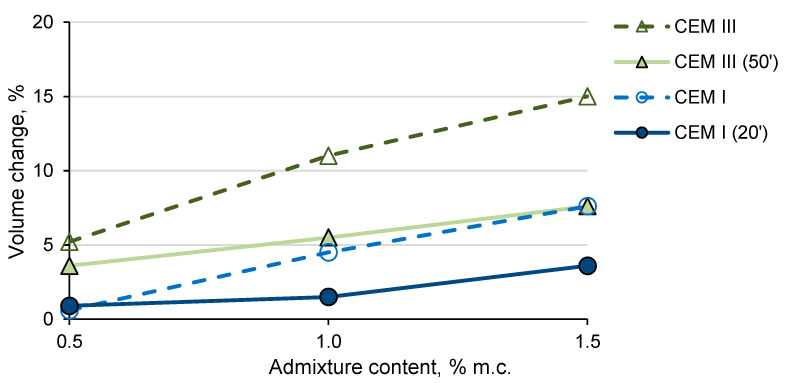
Change of mortar volume with the admixture in relation to the mortar without admixture formed after mixing components of mortar (CEM I and CEM III) and formed after air content test (CEM I (20′) and CEM III (50′)).

**Table 1 materials-14-02927-t001:** Tested mortars.

Mortar	CEM I 0%	CEM I 0.5%	CEM I 1.0%	CEM I 1.5%	CEM III 0%	CEM III 0.5%	CEM III 1.0%	CEM III 1.5%
admixture content, % m.c.	0	0.5	1.0	1.5	0	0.5	1.0	1.5
cement type	CEM I	CEM I	CEM I	CEM I	CEM III	CEM III	CEM III	CEM III

**Table 2 materials-14-02927-t002:** Increase in mortar sample height in relation to reference bars depending on the amount of gas-liberating admixture.

Admixture Content (% m.c.)	Increase in Height of Mortar Sample with CEM I (%)	Increase in Height of Mortar Sample with CEM III (%)
0.5	0.6	5.2
1.0	4.5	11
1.5	7.6	15

**Table 3 materials-14-02927-t003:** Increase in mortar sample height in relation to reference bars depending on the amount of gas-liberating admixture.

Admixture Content (% m.c.)	Bulk Density of Mortars with CEM I Cement (g/cm^3^)	Bulk Density of Mortars with CEM I Cement (g/cm^3^)
0.5	2.19 ± 0.02	2.25 ± 0.01
1.0	2.14 ± 0.01	2.14 ± 0.01
1.5	2.05 ± 0.01	2.03 ± 0.01

**Table 4 materials-14-02927-t004:** Parameters of pore structure calculated by the computer image analysis method.

Mortar	Relative Volume Fraction of Pores—*V*_V_ (%)	Relative Surface Area of Pores—*S*_V_ (mm^−1^)
CEM I 0%	7.17	0.66
CEM I 1.0%	11.1	1.10
CEM I 1.5%	17.4	1.73
CEM III 0%	4.24	0.37
CEM III 1.0%	8.98	1.21
CEM III 1.5%	18.4	2.22

**Table 5 materials-14-02927-t005:** Dependence of compressive strength on the amount of admixture containing aluminum powder and type of cement.

t, Days	Mortars with CEM I		Mortars with CEM III	
	*f*_c_ = f(*C*_A_)	R^2^	*f*_c_ = f(*C*_A_)	R^2^
7	*f*_c_ = −11.553*C*_A_ + 43.443 *f*_c_ * = −13.606*C*_A_ + 45.4838	0.979 0.999 *	*f*_c_ = −12.905*C*_A_ + 35.038	0.987
28	*f*_c_ = −10.065*C*_A_ + 47.947 *f*_c_ * = −12.771*C*_A_ + 51.104	0.9257 0.9525 *	*f*_c_ = −17.79*C*_A_ + 56.49	0.994
56	*f*_c_ = −13.285*C*_A_ + 52.097 *f*_c_ * = −18.417*C*_A_ + 58.083	0.8893 0.9685 *	*f*_c_ = −16.85*C*_A_ + 59.551	0.9714
91	*f*_c_ = −17.305*C*_A_ + 56.861 *f*_c_ * = −24.855*C*_A_ + 65.669	0.8866 0.9988 *	*f*_c_ = −19.178*C*_A_ + 61.853	0.9967

* dependence in the content of admixture range *C*_A_ є <0.5–1.5%>.

**Table 6 materials-14-02927-t006:** Dependence of flexural strength on the amount of admixture containing aluminum powder and type of cement.

t, Days	Mortars with CEM I		Mortars with CEM III	
	*f*_f_ = f(*C*_A_)	R^2^	*f*_f_ = f(*C*_A_)	R^2^
7	*f*_f_ = −2.585*C*_A_ +9.453 *f*_f_ * = −3.357*C*_A_ +10.353	0.924 0.9676	*f*_f_ = −2.031*C*_A_ +7.905	0.9939
28	*f*_f_ = −1.038*C*_A_ +9.574 *f*_f_ * = −1.081*C*_A_ +9.624	0.956 0.906	*f*_f_ = −3.304*C*_A_ +10.501	0.9795
56	*f*_f_ = −1.691*C*_A_ +8.916 *f*_f_ * = −2.205*C*_A_ +9.515	0.891 0.9184	*f*_f_ = −2.885*C*_A_ +11.006	0.963
91	*f*_f_ = −1.508*C*_A_ +8.702 *f*_f_ * = −1.889*C*_A_ +9.147	0.9563 0.9951	*f*_f_ = −2.121*C*_A_ +9.138	0.9062

* dependence in content of admixture range *C*_A_ є <0.5%–1.5%>.

## Data Availability

The data presented in this study are available on request from the corresponding author.

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
