# Peer review of "Influence of the Type of Cement on the Action of the Admixture Containing Aluminum Powder"

_materials, 2021, doi:10.3390/ma14112927_

Round 1
Reviewer 1 Report
The paper studies the influence of the gas liberating admixture containing aluminium powder on the properties of mortars with CEM I and CEM III cement.
The article shows some potential. However, there are some comments and suggestions:
The English language of the entire article must be revised because there are many grammar and spelling errors.
The introduction is poor in content and should be extended with a more extensive literature review.
Although Figures 6a and 6b are mentioned in the text at lines 247 and 263 they are missing from the manuscript.
There are numerous instructions for authors regarding the manuscript preparation, that should be deleted from the paper, such as:
- line 152: “All figures and tables should be cited in the main text as Figure 1, Table 1, etc.”;
- line 375, 376: “This section is not mandatory but can be added to the manuscript if the discussion is unusually long or complex.”
- line 377, 378: “Supplementary Materials: The following are available online at www.mdpi.com/xxx/s1, Figure S1: title, Table S1: title, Video S1: title.”;
- line 379: “data curation, X.X.”.
There are incomplete sentences at lines 239-240 (In the case of mortars) and at line 335 (In the case).
The values of flexural strength of the mortars with CEM III from the Table 6 (last column) are identical with the values of the compressive strength of the mortars with CEM III from the Table 5 (last column)?
The references section should be improved; several ISI papers would improve the scientific quality of the paper.
From these reasons, I think that this manuscript only after a major revision, could be accepted to be published by the Materials journal.
Author Response
Dear Reviewer,
Thank you very much for your good comments and suggestions concerning our manuscript. Those comments are all valuable and very helpful for revising and improving our paper. We have studied comments carefully and modified the manuscript accordingly. We hope they can meet your approval. The response to your comments is in author-coverletter-11944196.v1.pdf file.
Kind regards
Dr Justyna Kuziak

Reviewer 2 Report
Interesting results are reported and the manuscript was reviewed for publication in Materials-MDPI Journal. However, to warrant publication, the authors need to consider following major points -
- Introduction is well-written. However, more citable articles to cover the diversity of the subject are (a) https://doi.org/10.1016/j.conbuildmat.2015.07.122;
(b) https://doi.org/10.1016/j.cemconres.2005.12.017;
(c) https://doi.org/10.1016/j.cemconres.2013.03.016.
- How hardness and flexural strength of the concrete were estimated?
- How addition of aluminium powder effects the final properties (such as flexural strength) of different type of cement concrete used in this study?
- Figure 1 shows that air cavities were highest in case of CEM I after 12 minutes. How presence of higher air cavities affects final properties of concrete?
- In table-2, it can be noted that the expansion ratio that is increase in height of motor improves drastically in both CEM I and CEM III with increase in admixture content? Why? Also, the expansion was higher in CEM III than CEM I? why?
- From Figure 3, it can be seen that formation of cracks on motor sample were more on top surface than lateral surface? Is there any chemistry or particular reason for that?
- In Figure 7, the flexural strength was higher for CEM III and decrease with increase of admixture content? why?
- Please provide the conclusion in form of paragraph and NOT in form of bullets.
Author Response
Dear Reviewer,
Thank you very much for your good comments and suggestions concerning our manuscript. Those comments are all valuable and very helpful for revising and improving our paper. We have studied comments carefully and modified the manuscript accordingly. We hope they can meet your approval. The response to your comments is in author-coverletter-11947157.v1.pdf file.
Kind regards
Dr Justyna Kuziak

Round 2
Reviewer 1 Report
The paper has been revised according to the reviewers’ comments, and the authors tried to make all the corrections according to these suggestions.
However, in the author response “Response to comments of reviewer #1.” It is said that the data in Table 5 (Table 5. Dependence of compressive strength on the amount of admixture containing aluminium powder and type of cement) has been corrected (as shown below…), but in the manuscript this correction was made in Table 6. (Table 6. Dependence of flexural strength on the amount of admixture containing aluminium powder and type of cement).
Which of the two versions is correct?
From these reasons, I think that this manuscript, after a minor revision, could be accepted to be published by the Materials journal.
Author Response
Dear Reviewer,
Thank you for reading our article and response to your comments carefully. In the last response “Response to comments of reviewer #1” I gave the wrong number and title of the table. There should be table 6 and flexural strength instead of compressive strength in the title (Table 6 was presented in the response, but with the wrong number and title). I apologize for my inattention. The content of Table 5 and Table 6 in the manuscript was once again verified. Correct version was in the manuscript (the correct table (Table 6) was corrected the previous time).
We sincerely appreciate your work and hope that our manuscript meets with your approval.
Kind regards
Dr Justyna Kuziak
Reviewer 2 Report
I have no further comments.
Author Response
Dear Reviewer,
Thank you for your opinion.
We appreciate your work and believe that our manuscript has received your approval.
Kind regards
Dr. Justyna Kuziak